

# Future $CO_2$-induced seawater acidification mediates the physiological performance of a green alga *Ulva linza* in different photoperiods

Furong Yue[1], Guang Gao[1], Jing Ma[1], Hailong Wu[1], Xinshu Li[1] and Juntian Xu[1,2,3]

[1] Jiangsu Key Laboratory of Marine Bioresources and Environment, Huaihai Institute of Technology, Lianyungang, China
[2] Co-Innovation Center of Jiangsu Marine Bio-industry Technology, HuaiHai Institute of Technology, Lianyungang, China
[3] Jiangsu Key Laboratory of Marine Biotechnology, Huaihai Institute of Technology, Lianyungang, China

Corresponding author
Juntian Xu, jtxu@hhit.edu.cn

## ABSTRACT

Photoperiods have an important impact on macroalgae living in the intertidal zone. Ocean acidification also influences the physiology of macroalgae. However, little is known about the interaction between ocean acidification and photoperiod on macroalgae. In this study, a green alga *Ulva linza* was cultured under three different photoperiods (L: D = 8:16, 12:12, 16:8) and two different $CO_2$ levels (LC, 400 ppm; HC, 1,000 ppm) to investigate their responses. The results showed that relative growth rate of *U. linza* increased with extended light periods under LC but decreased at HC when exposed to the longest light period of 16 h compared to 12 h. Higher $CO_2$ levels enhanced the relative growth rate at a L: D of 8:16, had no effect at 12:12 but reduced RGR at 16:8. At LC, the L: D of 16:8 significantly stimulated maximum quantum yield (Yield). Higher $CO_2$ levels enhanced Yield at L: D of 12:12 and 8:16, had negative effect at 16:8. Non-photochemical quenching (NPQ) increased with increasing light period. High $CO_2$ levels did not affect respiration rate during shorter light periods but enhanced it at a light period of 16 h. Longer light periods had negative effects on Chl *a* and Chl *b* content, and high $CO_2$ level also inhibited the synthesis of these pigments. Our data demonstrate the interactive effects of $CO_2$ and photoperiod on the physiological characteristics of the green tide macroalga *Ulva linza* and indicate that future ocean acidification may hinder the stimulatory effect of long light periods on growth of *Ulva* species.

## INTRODUCTION

Due to the activities of humans, the concentration of atmospheric $CO_2$ has increased to 400 ppm from 278 ppm during the pre-industrial revolution (*Gattuso et al., 2015*). It is estimated that the oceans, as a $CO_2$ sink, have taken up approximately 48% of the fossil-fuel and cement-manufacturing emissions (*Sabine et al., 2004*). The concentration of atmospheric $CO_2$ has been predicted to reach almost 1,000 ppm by the end of 21st

century (*Gattuso et al., 2015*). This would result in the decline of surface ocean pH by 0.3–0.4 units, an increase of hydrogen ion concentration and bicarbonate concentration in seawater, which is termed ocean acidification (OA) (*Orr et al., 2005*; *Koch et al., 2013*; *Albright, 2016*). OA will be maintained for hundreds of years, and thus could cause a huge change in marine ecosystems (*Kroeker et al., 2013*; *Wittmann & Pörtner, 2013*).

Marine phytoplankton accounts for 50% of global primary production (*Beardall, Stojkovic & Larsen, 2009*) and macroalgae accounts for 10% of marine primary productivity (*Smith, 1981*; *Charpy-Roubaud & Sournia, 1990*; *Heinrich et al., 2012*). Macroalgae have evolved distinct strategies to capture inorganic carbon (Ci) for photosynthesis: (1) by the diffusive uptake of $CO_2$ (*Giordano, Beardall & Raven, 2005*; *Raven et al., 2005*), (2) the active transport of $HCO_3^-$/$CO_2$ (termed carbon concentrating mechanisms (CCMs)) (*Maberly, Raven & Johnston, 1992*; *Raven, 2003*; *Giordano, Beardall & Raven, 2005*). The elevated $CO_2$ might enhance the growth and photosynthesis of algae that depend on $CO_2$ diffusion or have weak CCMs (*Kübler, Johnston & Raven, 1999*; *Gao et al., 2016a*). Therefore, the physiological influences of OA on macroalgae might be species-specific. For example, OA has a positive influences on the growth of macroalgae, such as the red algae *Pyropia yezoensis, Pyropia haitanensis* and *Gracilaria lemaneiformis* (*Gao et al., 1991*; *Xu & Gao, 2015*; *Chen et al., 2018*), green algae *Ulva prolifera, Ulva lactuca, Caulerpa taxifolia* (*Xu & Gao, 2012*; *Chen, Zou & Jiang, 2015*; *Roth-Schulze et al., 2018*) and brown algae *Hizikia fusiforme, Ecklonia radiata, Sargassum muticum* (*Zou, 2005*; *Britton et al., 2016*; *Xu et al., 2017*). However, research has shown that high $CO_2$ levels did not have a detectable effect on the relative growth rate of several species, for example, *Sargassum henslowianum, Ulva rigida, Ulva australis*, the giant kelp *Macrocystis pyrifera* and *Ulva australis* (*Chen & Zou, 2014*; *Rautenberger et al., 2015*; *Fernández, Roleda & Hurd, 2015*; *Reidenbach et al., 2017*). Furthermore, increasing seawater $CO_2$ concentration reduced the growth of *Gracilaria tenuistipitata*, **Porphyra leucosticte**, *Fucus vesiculosus, Halimeda opuntia* and *Ulva linza* (*García-Sánchez, Fernández & Niell, 1994*; *Mercado et al., 1999*; *Gutow et al., 2014*; *Johnson, Price & Smith, 2014*; *Gao et al., 2018*). In addition, the various effects of ocean acidification may be also modulated by other environmental factors, such as temperature, nutrients and photoperiods (*Gao et al., 2017a*; *Gao et al., 2019*; *Li et al., 2018*).

Photoperiod is a key factor regulating seasonal responses of algae (*Dring, 1988*). Furthermore, daylength can influence algal affinity for dissolved inorganic carbon and cellular carbon demand, thus imposing an effect on CCMs regulation (*Rost, Riebesell & Sültemeyer, 2006*). Thus, the photoperiod may be critical for algae cultured under elevated $CO_2$ levels. Meanwhile, the interactive effects of OA and increased photoperiods could enhance the growth of *U. prolifera*, thus increase the opportunities for occurrence of green tides (*Li et al., 2018*). Elevated $CO_2$ levels enhanced the photosynthetic rate, but longer illumination periods reduced the photosynthetic efficiency in *E. huxleyi* (*Nielsen, 1997*; *Rost, Riebesell & Sültemeyer, 2006*). Therefore, at HC, the responses of algae to different photoperiods appear to be genera- and species-specific.

Green macroalgal bloom has occurred each year in the Yellow Sea (YS) since 2007, termed green tides (*Xu, Zhang & Cheng, 2016*). Green tides generate aesthetic problems and toxic gas when thalli decompose (*Gao et al., 2016b*). *Ulva* is the dominant genus

contributing majority of green tides and *U. linza* is one of species causing green tides in the Yellow Sea (*Fletcher, 1996*; *Kang et al., 2014*; *Gao et al., 2017a*; *Gao et al., 2017b*). However, little is known about the interactive effects of ocean acidification and photoperiod on *U. linza*. Previous study showed that the effect of ocean acidification on diatoms was related to light intensity (*Gao et al., 2012*). In this study, based on previous studies, we hypothesized the effect of ocean acidification on *U. linza* may dependent on photoperiod. To test our hypothesis, the physiological responses of the green macroalga *Ulva linza* under two different $CO_2$ levels and three different photoperiods were examined.

## MATERIALS & METHODS

### Thalli collection and culture conditions

*U. linza* was collected from the coastal water of Gaogong peninsula (119.3°E, 34.5°N), Lianyungang, Jiangsu Province, China. Gaogong peninsula is a public place and no approval is required for collecting naturally growing *Ulva* species in China because *Ulva* species cause green tides in coastal waters of the Yellow Sea. *U. linza* was identified by morphological characters (*Ma et al., 2009*). The samples were transferred to the laboratory in a portable cooler (4–6 °C) box within one hour. Healthy thalli (first observed by color and then checked with maximum quantum yield of PSII) were selected and cleaned with filtered natural seawater to remove sediments, visible epiphytes and attached animals. Thalli were pre-cultured in a 500 ml flask in an illuminated incubator (GXZ-500B, Ningbo, China) at 20 °C, with the illumination intensity set at 150 $\mu$mol photons m$^{-2}$s$^{-1}$ (12L:12D). Sterilized filtered seawater (salinity 30, supplied with 8 $\mu$M $NaH_2PO_4$ and 60 $\mu$M $NaNO_3$) was used as culture medium and the medium was bubbled with air before being renewed every two days.

For these experiments, approximately three 1 cm long thalli were placed into a 550 ml flask filled with 500 ml sterile seawater aerated ambient outdoor air (400 ppm, LC) or with $CO_2$-enriched air (1,000 ppm, HC) and randomly cultured in three separate incubators with different light levels (8L:16D, 12L:12D, 16L:8D) and continuous aeration at 20 °C with 150 $\mu$mol photons m$^{-2}$s$^{-1}$. Different $CO_2$ levels (400 ppm, LC; 1,000 ppm, HC) were achieved by bubbling ambient air or $CO_2$-enriched air via a $CO_2$ plant incubator (HP 1000 G-D, Ruihua Instruments, Wuhan, China).

To maintain the pH$_{NBS}$ at about 8.12 (LC) and 7.78 (HC) under different photoperiods, the increased algal biomass were removed constantly and the medium was changed every two days and daily variations in pH were maintained at less than 0.05. The experiment lasted 9 days, the physiological factors were taken on the last 3 days.

### Estimate of carbonate system parameters

The seawater pH was monitored with a pH meter (pH 700, Eutech Instruments, Singapore) and total alkalinity (TA) was calculated by titrations (*Gao et al., 2019*). Other parameters of the carbonate system were obtained with CO2SYS software (*Pierrot, Lewis & Wallace, 2006*), the equilibrium constants $K_1$ and $K_2$ for carbonic acid dissociation (*Roy et al., 1993*).

## Measurement of growth

The length of *U. linza* was recorded every 2 days. The relative growth rate (RGR) of thalli were calculated across the 6-day period before other physiological parameters were measured during the following 3 days. RGR was calculated as follows: $RGR = \ln (W_t/W_0)/t$, where $W_t$ is the length after t days culture, $W_0$ is the initial length.

## Chlorophyll fluorescence measurements

Chlorophyll *a* fluorescence in *U. linza* was measured with a portable PAM (Pulse-amplitude-modulation; AquaPen-P AP-P 100, Chech). Algae were dark adapted for 15 min before the experiment. The parameters were calculated according to the following equations:

$rETR = PAR \times Y(II) \times 0.84 \times 0.5$ (*Schreiber, 2004*; *Zhang, Zhang & Yang, 2017*),

where rETR is the relative electron transport rate; PAR is the actinic light; Y(II) is the effective quantum yield of PSII.

$NPQ = Fm - Fm'/1$ (*Bilger & Schreiber, 1986*),

where NPQ is the non-photochemical quenching; Fm is the maximum fluorescence value of *U. linza* when they were adapted in the dark for 15 min; Fm' is the maximum fluorescence value of *U. linza* under actinic light conditions.

## Respiration rate measurements

The respiration rate of *U. linza* was measured using a Clark-type oxygen electrode (YSI Model 5300, Yellow Springs Instrument Co., USA). The thallus was cut into 1 cm long segments with scissors and the thalli were placed in culture conditions for at least 1 h to decrease the effects of cutting damage. Approximately 0.01 g fresh weight of thalli were placed in the reaction chamber with 8 ml medium. Temperature was controlled at 20 °C with a circulating water bath. The decrease of the oxygen content in the seawater in darkness with seven minutes was defined as the respiration rate.

## Measurement of photosynthetic pigments

Chlorophyll *a* and *b* were extracted from thalli (about 10 mg FW) with 5 ml methanol at 4 °C for 24 h in the dark. The absorption values were obtained at 652 nm, 663 nm and 665 nm using an ultraviolet spectrophotometer (Ultrospect 3300 pro; Amersham Bioscience, Sweden). The contents of the Chl *a* and Chl *b* were estimated using the method of *Porra, Thompson & Kriedemann (1989)*.

## Data analysis

All the data are shown as mean ± SD. Origin 9.0 and SPSS 18.0 were used to analyze data. Two-way ANOVA was used to assess the interactive effects of $CO_2$ levels and photoperiods on relative growth rate, chlorophyll fluorescence parameters, respiration rate and pigment content of *U. linza*. One-way ANOVA was used to analyze differences under the same conditions. Tukey HSD was conducted for *post hoc* investigation. Confidence intervals were set at 95%.

**Table 1 Parameters of the seawater carbonate system in different cultures.**

|  | $pH_{NBS}$ | $p\,CO_2$ ($\mu$atm) | DIC ($\mu$mol kg$^{-1}$) | $HCO_3^-$ ($\mu$mol kg$^{-1}$) | $CO_3^{2-}$ ($\mu$mol kg$^{-1}$) | $CO_2$ ($\mu$mol kg$^{-1}$) | TA ($\mu$mol kg$^{-1}$) |
|---|---|---|---|---|---|---|---|
| 8:16 LC | 8.14 ± 0.02[a] | 580.32 ± 35.74[a] | 2597.04 ± 57[a] | 2385.19 ± 56.15[a] | 192.59 ± 7.23[a] | 19.56 ± 1.19[a] | 2850.78 ± 56.26[a] |
| 12:12 LC | 8.12 ± 0.01[a] | 613.13 ± 11.21[a] | 2594.32 ± 61.96[a] | 2391.03 ± 55.85[a] | 182.94 ± 6.07[a] | 20.35 ± 0.37[a] | 2834.31 ± 67.99[a] |
| 16:8 LC | 8.13 ± 0.01[a] | 608.54 ± 14.83[a] | 2638.26 ± 40.24[a] | 2427.94 ± 35.09[a] | 190.12 ± 6.73[a] | 20.19 ± 0.49[a] | 2886.86 ± 46.67[a] |
| 8:16 HC | 7.74 ± 0.03[b] | 1614.77 ± 100.05[b] | 2780.28 ± 12.05[b] | 2641.71 ± 12.23[b] | 84.99 ± 5.08[b] | 53.58 ± 3.32[b] | 2854.16 ± 13.32[a] |
| 12:12 HC | 7.79 ± 0.02[cd] | 1445.67 ± 58.75[c] | 2799.47 ± 14.61[b] | 2655.70 ± 13.74[b] | 95.80 ± 3.81[b] | 47.97 ± 1.95[c] | 2893.15 ± 17.18[a] |
| 16:8 HC | 7.75 ± 0.01[bd] | 1652.45 ± 43.38[b] | 2938.18 ± 47.63[c] | 2790.86 ± 45.52[c] | 92.48 ± 1.28[b] | 54.83 ± 1.44[b] | 3019.14 ± 46.95[b] |

Data are the mean ± SD ($n = 3$).
DIC, dissolved inorganic carbon; TA, total alkalinity.

# RESULTS

Both the elevated $CO_2$ levels and photoperiod altered carbonate parameters in seawater, and they both had an interactive effect (Tables 1 and 2). The elevated $CO_2$ decreased pH and $CO_3^{2-}$, increased $p\,CO_2$, DIC, $HCO_3^-$ and $CO_2$ in the seawater. Increased photoperiod did not affect carbonate parameters at LC but the longest photoperiod increased DIC, $HCO_3^-$ and TA compared to shortest photoperiod.

The two-way ANOVA showed that elevated $CO_2$ and photoperiod had an interactive effect, and both elevated $CO_2$ levels and the photoperiods had a significant effect on the RGR of *U. linza* (Fig. 1 and Table 3). At LC, the RGR of adult *U. linza* increased with the extended light periods, and the highest RGR occurred at a L: D of 16:8. The effect of $CO_2$ also varied with photoperiod. Higher $CO_2$ levels enhanced RGR at L: D of 8:16, but had no effect at 12:12 and reduced it at 16:8.

The Yield and NPQ were measured under different $CO_2$ levels and photoperiod conditions (Fig. 2). Two-way ANOVA showed that elevated $CO_2$ and photoperiod had an interactive effect on Yield (Table 4). Higher $CO_2$ levels increased Yield when thalli were cultured under photoperiods of 8:16 and 12:12 but reduced it under 16:8. Photoperiod had the main effect on NPQ (Table 4). At LC, thalli cultured at L: D of 16:8 had higher NPQ compared to L: D of 8:16 while the difference between 8:16 and 12:12 was insignificant. At HC, NPQ increased with the increase in photoperiod although the increase was not statistically significant. The elevated $CO_2$ had neutral effect on NPQ of *U. linza*.

Maximum rETR (rETRmax), efficiency of electron transport ($\alpha$), and saturating irradiance ($I_k$) were calculated from the rapid light curves (Fig. 3, Table 5). Photoperiod and elevated $CO_2$ levels had an interactive effect, and elevated $CO_2$ levels had a main effect on light-saturated electron transport rate (rETRmax) (Table 6). Higher $CO_2$ levels increased rETRmax during the 8:16 and 12:12 photoperiods, but did not affect it at 16:8. A similar pattern was also found for $\alpha$. In contrast to rETRmax and $\alpha$, $CO_2$ did not affect $I_k$ while photoperiod had the main effect on it. At LC, $I_k$ increased when L: D rose from 8:16 to 12:12 but did not change with the further increase in photoperiod. At HC, $I_k$ did not change when L: D rose from 8:16 to 12:12 but was enhanced when L: D increased to 16:8 (Table 5).

Yue et al. (2019), *PeerJ*, DOI 10.7717/peerj.7048

**Table 2** Two-way ANOVA analysis of variance for the effects of $CO_2$ and photoperiod regimes on pH, $p\,CO_2$, dissolved inorganic carbon (DIC), $HCO_3^-$, $CO_3^{2-}$, $CO_2$ and total alkalinity (TA) in seawater.

| Source | df | pH | | $p\,CO_2$ | | DIC | | $HCO_3^-$ | | $CO_3^{2-}$ | | $CO_2$ | | TA | |
|---|---|---|---|---|---|---|---|---|---|---|---|---|---|---|---|
| | | F | Sig. | F | Sig. | F | Sig. | F | Sig. | F | Sig. | F | Sig. | F | Sig. |
| Photoperiod | 2 | 1.333 | 0.3 | 5.640 | 0.019 | 9.697 | 0.003 | 10.112 | 0.003 | .353 | 0.710 | 5.632 | 0.019 | 8.638 | 0.005 |
| $CO_2$ | 1 | 2268.750 | <0.001 | 1498.294 | <0.001 | 125.390 | <0.001 | 236.530 | <0.001 | 1453.669 | <0.001 | 1497.372 | <0.001 | 8.977 | 0.011 |
| Photoperiod*$CO_2$ | 2 | 7.750 | 0.007 | 7.562 | 0.007 | 3.054 | 0.085 | 3.184 | 0.078 | 5.342 | 0.022 | 7.557 | 0.008 | 2.976 | 0.089 |
| Error | 12 | | | | | | | | | | | | | | |

**Notes.**

$CO_2$*photoperiod means the interactive effects of $CO_2$ and photoperiod, df mean degree of freedom and F means the value of F statistic, and Sig. means *p*-value.

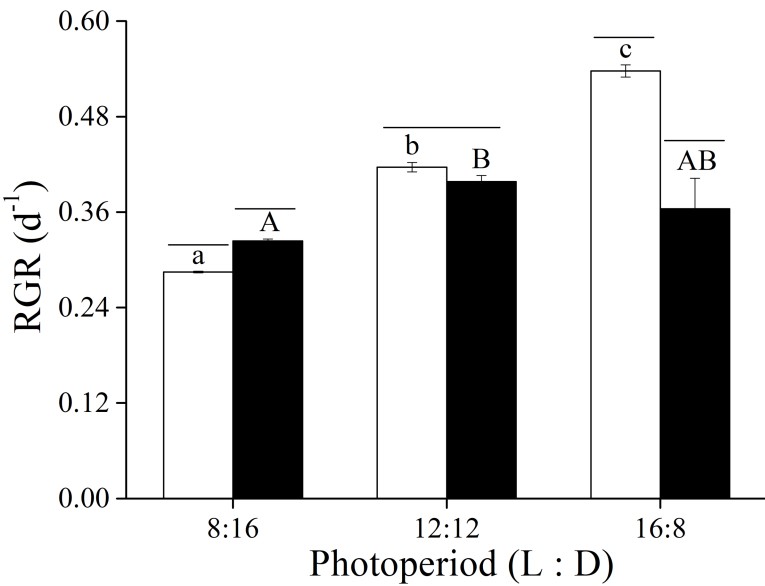

**Figure 1** **Relative growth rate.** Relative growth rate (RGR) of *U. linza* thalli grown at different $CO_2$ levels (400 ppm, LC; 1000 ppm, HC) and different photoperiods (L: D = 8:16, 12:12, 16:8).

**Table 3** **Two-way ANOVA analysis of variance for the effects of $CO_2$ and photoperiod regimes on RGR of *Ulva linza* cultured at different $CO_2$ levels and photoperiod regimes.**

| Source | df | F | Sig. |
|---|---|---|---|
| Photoperiod | 2 | 124.249 | <0.001 |
| $CO_2$ | 1 | 42.292 | <0.001 |
| Photoperiod* $CO_2$ | 2 | 65.959 | <0.001 |
| Error | 12 | | |

**Notes.**
$CO_2$*photoperiod means the interactive effects of $CO_2$ and photoperiod, df mean degree of freedom and F means the value of F statistic, and Sig. means *p*-value.

In addition to photosynthetic parameters, the effects of $p$ $CO_2$ and photoperiod on the respiration rate of adult *U. linza* were also investigated (Fig. 4). Photoperiod and elevated $CO_2$ levels had an interactive effect, and both photoperiod and elevated $CO_2$ levels had the primary effect on the respiration rate of *U. linza* (Table 7). Higher $CO_2$ levels did not affect the respiration rate at photoperiods of 8:16 or 12:12 but increased it by 56.39% at a photoperiod of 16:8.

Changes in photosynthetic pigments of *U. linza* grown under various conditions are shown in Fig. 5. Two-way ANOVA showed that $CO_2$ and photoperiod had an interactive effect, and both $CO_2$ levels and the photoperiods had the main effect on the Chl *a* content of *U. linza* (Table 8). Prolonged light periods reduced the synthesis of Chl *a* in thalli although the difference between the photoperiods of 8:16 and 12:12 at HC was not statistically significant. Higher $CO_2$ levels reduced Chl *a* at the photoperiods of 8:16 and 12:12 but did not affect it at a L: D of 12:12. The same trend was found for Chl *b*. The Chl *a/b* ratios were all greater than 1.0, suggesting a higher synthesis of Chl *a* than Chl *b* under all culture
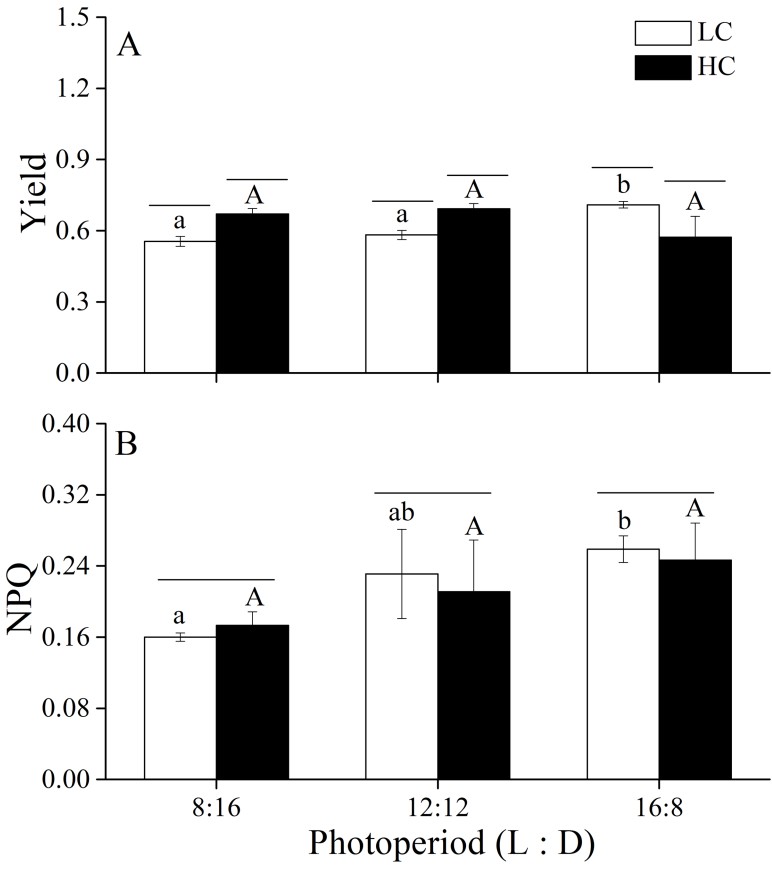

**Figure 2 Yield and NPQ.** Yield (A) and NPQ (B) of *U. linza* thalli grown at different $CO_2$ levels (400 ppm, LC; 1000 ppm, HC) and different photoperiods (L: D = 8:16, 12:12, 16:8).

**Table 4 Two-way ANOVA analysis of variance for the effects of $CO_2$ and photoperiod regimes on Yield and NPQ of *Ulva linza* cultured at different $CO_2$ levels and photoperiod regimes.**

| Source | Yield | | | NPQ | | |
|---|---|---|---|---|---|---|
| | df | F | Sig. | df | F | Sig. |
| Photoperiod | 2 | 0.919 | 0.425 | 2 | 8.453 | 0.005 |
| $CO_2$ | 1 | 2.549 | 0.136 | 1 | 0.132 | 0.722 |
| Photoperiod* $CO_2$ | 2 | 19.786 | <0.001 | 2 | 0.339 | 0.719 |
| Error | 12 | | | 12 | | |

**Notes.**
$CO_2$*photoperiod means the interactive effects of $CO_2$ and photoperiod, df mean degree of freedom and F means the value of F statistic, and Sig. means *p*-value.

conditions. Photoperiod and elevated $CO_2$ levels had an interactive effect on the Chl *a/b* ratio (Table 8); the higher $CO_2$ levels increased the ratio at a photoperiod of 12:12 but not at the other photoperiods.

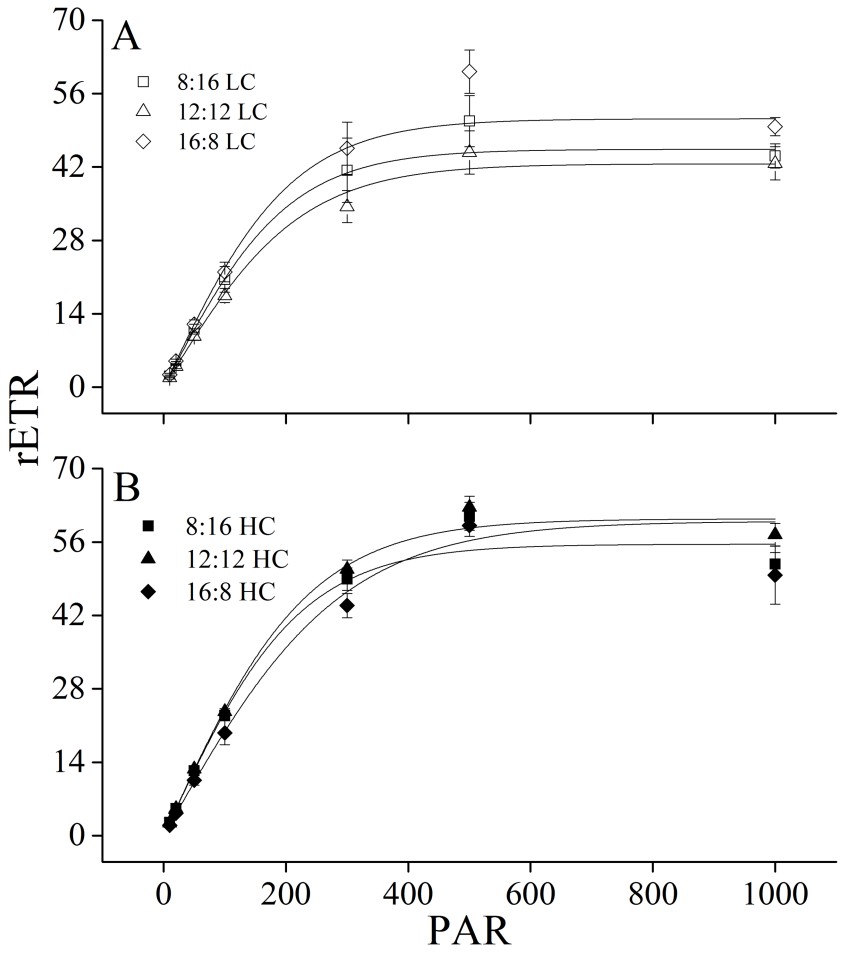

**Figure 3** **Rapid light curves.** Rapid light curves (RLC) of *U. linza* thalli grown at different $CO_2$ levels (400 ppm, LC; 1,000 ppm, HC) and different photoperiods (L: D = 8:16, 12:12, 16:8).

**Table 5** **Parameters of rapid light curves (RLC) of *U. linza* thalli cultured at different $CO_2$ levels and photoperiod regimes.**

|          | rETRmax                  | $\alpha$              | $I_k$                      |
|----------|--------------------------|-----------------------|----------------------------|
| 8:16 LC  | $47.53 \pm 3.63^a$       | $0.22 \pm 0.04^a$     | $215.67 \pm 19.89^a$       |
| 12:12 LC | $43.87 \pm 3.60^c$       | $0.18 \pm 0.02^c$     | $248.87 \pm 22.91^{bd}$    |
| 16:8 LC  | $54.83 \pm 2.65^b$       | $0.24 \pm 0.03^{ab}$  | $231.58 \pm 21.37^{adef}$  |
| 8:16 HC  | $56.51 \pm 2.34^b$       | $0.25 \pm 0.01^b$     | $222.15 \pm 13.41^{ac}$    |
| 12:12 HC | $60.45 \pm 1.38^d$       | $0.26 \pm 0.01^b$     | $236.58 \pm 12.63^{bce}$   |
| 16:8 HC  | $54.55 \pm 3.77^b$       | $0.22 \pm 0.01^a$     | $251.72 \pm 11.69^{bf}$    |

Data are the mean $\pm$ SD ($n \geq 3$). Different letters represent significant difference ($p < 0.05$) among different treatments.

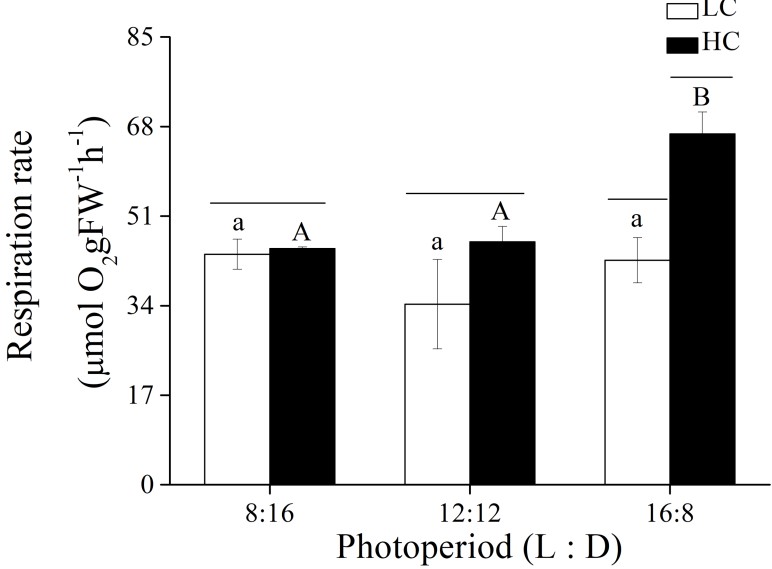

**Figure 4 Respiration rate.** Respiration rate of *U. linza* thalli grown at different $CO_2$ levels (400 ppm, LC; 1,000 ppm, HC) and different photoperiods (L: D = 8:16, 12:12, 16:8).

**Table 6 Two-way ANOVA analysis of variance for the effects of $CO_2$ and photoperiod regimes on maximum rETR (rETRmax), efficiency of electron transport ($\alpha$), and saturating irradiance ($I_k$) of *Ulva linza* cultured at different $CO_2$ levels and photoperiod regimes.**

| Source | rETRmax | | | $\alpha$ | | | $I_k$ | | |
|---|---|---|---|---|---|---|---|---|---|
| | df | F | Sig. | df | F | Sig. | df | F | Sig. |
| Photoperiod | 2 | 2.981 | 0.066 | 2 | 2.909 | 0.070 | 2 | 7.030 | 0.003 |
| $CO_2$ | 1 | 69.993 | <0.001 | 1 | 15.225 | <0.001 | 1 | 0.664 | 0.421 |
| Photoperiod* $CO_2$ | 2 | 23.445 | <0.001 | 2 | 14.896 | <0.001 | 2 | 2.576 | 0.093 |
| Error | 30 | | | 30 | | | 30 | | |

**Notes.**
$CO_2$*photoperiod means the interactive effects of $CO_2$ and photoperiod, df mean degree of freedom and F means the value of F statistic, and Sig. means *p*-value.

**Table 7 Two-way ANOVA analysis of variance for the effects of $CO_2$ and photoperiod regimes on respiration rate of *Ulva linza* cultured at different $CO_2$ levels and photoperiod regimes.**

| Source | df | F | Sig. |
|---|---|---|---|
| Photoperiod | 2 | 16.006 | <0.001 |
| $CO_2$ | 1 | 33.058 | <0.001 |
| Photoperiod* $CO_2$ | 2 | 9.566 | 0.003 |
| Error | 12 | | |

**Notes.**
$CO_2$*photoperiod means the interactive effects of $CO_2$ and photoperiod, df mean degree of freedom and F means the value of F statistic, and Sig. means *p*-value.

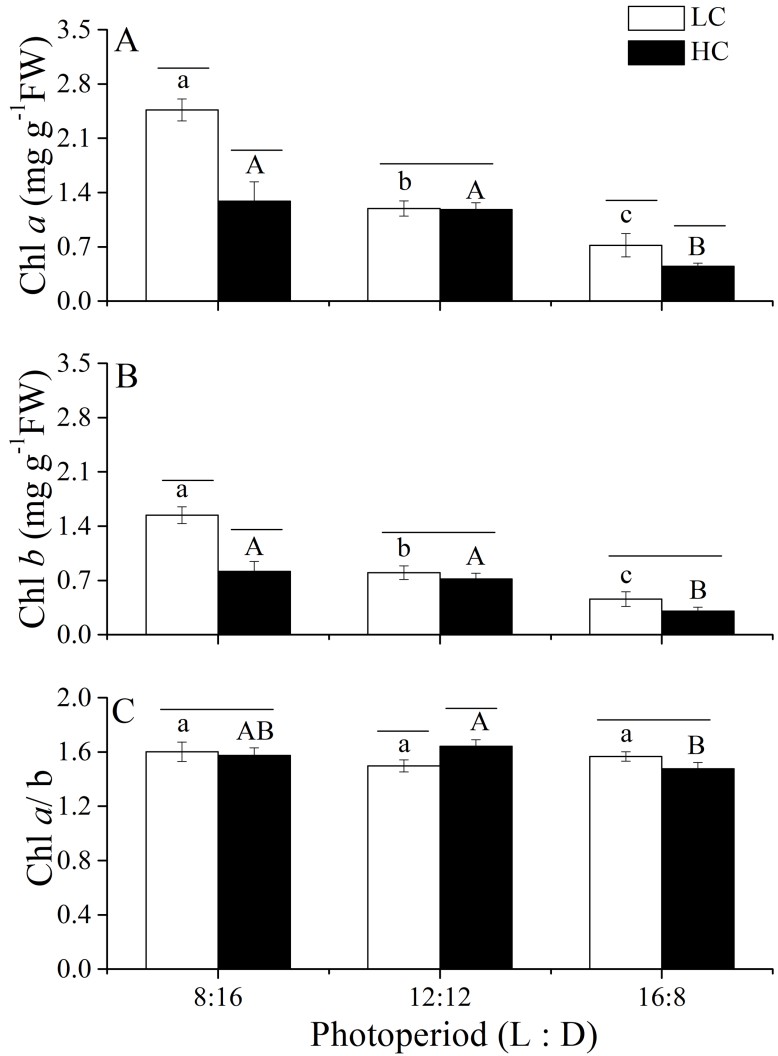

**Figure 5** **Pigment contents and the Chl a./b ratio.** Pigment contents (Chl *a*, (A); Chl *b*, (B)) and the Chl *a/b* ratio (C) of *U. linza* thalli grown at different $CO_2$ levels (400 ppm, LC; 1,000 ppm, HC) and different photoperiods (L: D = 8:16, 12:12, 16:8).

## DISCUSSION

In the present study, at LC, extended photoperiods had a positive effect on the relative growth rate of adult *U. linza,* similar to previous studies on *Laminaria sacharina, Porphyra umbilicalis* and *Ulva prolifera* (*Fortes & Lüning, 1980*; *Green & Neefus, 2016*; *Li et al., 2018*). Carbon isotope fractionation experiments suggested that extended photoperiods could enhance growth by influencing inorganic carbon capture and fixation rate in algae (*Rost et al., 2003*). This hypothesis is supported by the present study where extended photoperiods increased the maximum quantum yield in PS II. On the other hand, the highest growth rates of *Compsopogon coeruleus* were obtained in shorter light periods (L: D = 8:16) (*Zucchi & Necchi, 2001*), the highest growth rates of *Porphyra umbilicalis* was found under neutral

**Table 8** Two-way ANOVA analysis of variance for the effects of $CO_2$ and photoperiod regimes on Chl *a*, Chl *b*, and Chl *a/b* of *Ulva linza* cultured at different $CO_2$ levels and photoperiod regimes.

| Source | Chl *a* | | | Chl *b* | | | Chl *a/b* | | |
|---|---|---|---|---|---|---|---|---|---|
| | df | F | Sig. | df | F | Sig. | df | F | Sig. |
| Photoperiod | 2 | 121.283 | <0.001 | 2 | 111.035 | <0.001 | 2 | 2.625 | 0.113 |
| $CO_2$ | 1 | 51.793 | <0.001 | 1 | 53.849 | <0.001 | 1 | 0.171 | 0.686 |
| Photoperiod* $CO_2$ | 2 | 26.932 | <0.001 | 2 | 21.719 | <0.001 | 2 | 8.389 | 0.005 |
| Error | 12 | | | 12 | | | 12 | | |

**Notes.**

$CO_2$*photoperiod means the interactive effects of $CO_2$ and photoperiod, df mean degree of freedom and F means the value of F statistic, and Sig. means *p*-value.

(L: D = 12:12) and longer (L: D = 16:8) light periods using 110 µmol photons m$^{-2}$s$^{-1}$ (*Green & Neefus, 2016*), and the relative growth rate of *Chlorella vulgaris* cultured at L: D = 20:4 was lower than the RGR of alga cultured at L: D = 16:8 (*Kendirlioglu, Agirman & Cetin, 2015*). Therefore, the effects of photoperiod on algae appear to be species-specific.

Although *Ulva* has efficient mechanisms for $CO_2$ concentration, the growth of *Ulva* can be enhanced by elevated $CO_2$ (*Young & Gobler, 2016*; *Gao et al., 2016a*; *Gao et al., 2017a*; *Gao et al., 2017b*). However, in this study, we found that the effects of elevated $CO_2$ levels on the relative growth rate of *U. linza* depended on photoperiod. $CO_2$ enhanced the growth of adult *U. linza* under light/dark conditions of 8:16, had no effect on growth of *U. linza* under a L: D of 12:12, and reduced the relative growth rate under a L: D of 16:8. High $CO_2$ levels can down-regulate algal $CO_2$ concentration mechanisms (CCMs), meaning that energy would be saved, and thus enhancing the relative growth rate of algae (*Gao et al., 2012*; *Gao et al., 2016a*; *Raven, Beardall & Sánchez-Baracaldo, 2017*). This is supported by decreased pigment synthesis in thalli at higher $CO_2$ levels. However, higher $CO_2$ levels did not affect the growth rate of *U. linza* at medium photoperiods. The neutral effects of $CO_2$ on the growth of *U. rigida* (*Rautenberger et al., 2015*) and *U. linza* (*Gao et al., 2018*) were also reported. We speculate that this is a compromise between the positive effects of elevated $CO_2$ and negative effects of decreased pH. The negative effect of decreased pH on growth was documented for the brown alga *Sargassum muticum* (*Xu et al., 2017*). Algae might need to consume additional energy to act against acid–based perturbation caused by decreased pH, leading to reduced growth (*Xu et al., 2017*). This is supported by an enhanced respiration rate at the higher $CO_2$ levels in this study. The phenomenon of an increased respiration rate of algae under elevated $CO_2$ concentrations was found in *Hizikia fusiformis* (*Zou, 2005*), the microalgae *Phaeodactylum tricornutum* (*Wu, Gao & Riebesell, 2010*) and *Emiliania huxleyi* (*Jin et al., 2015*).

Furthermore, the higher $CO_2$ level reduced the growth rate of *U. linza* under the longest photoperiod in the present study. This may be due to the combination of down-regulated CCMs and excess light energy. The operation of CCMs is an energy-consuming process and the down-regulation of CCMs can result in additional energy (*Raven, Beardall & Sánchez-Baracaldo, 2017*). Higher light can usually reduce algal photosynthetic activity (*Singh & Singh, 2015*; *Gao et al., 2016a*). The energy saved due to down-regulation of CCMs at HC combined with high light intensity could synergistically damage the algal

photosystem and photosynthetic rate (*Gao et al., 2012*; *Gao et al., 2016a*). Although light intensity did not change among different $CO_2$ treatments, the lengthened photoperiod may have similar effect to increased light intensity. This argument is supported by increased NPQ at longer photoperiods as NPQ is photo-protective process to dissipate excess light energy. Higher $CO_2$ levels also reduced maximum quantum yield and stimulated the respiration rate of *U. linza* during the longest photoperiod in this study, leading to the decrease in growth. Enhanced respiration rate is a signal that organisms are fighting against stress and damage (*Xu et al., 2017*).

In addition to growth, the interactive effect of $CO_2$ and photoperiod was also found in photosynthetic parameters. For instance, elevated $CO_2$ increased Yield, rETRmax and $\alpha$ at shorter photoperiods (L: D of 8:16 and 12:12) but did not affect rETRmax and reduced Yield at longest photoperiod (L: D of 16:8). These findings indicate the close connection between growth and photosynthesis in terms of responding to the combination of $CO_2$ and photoperiod. It is worth noting that the interaction of $CO_2$ and photoperiod on growth of *U. linza* in this study is different from the findings in *U. prolifera* reported by *Li et al. (2018)*. Elevated $CO_2$ increased growth of *U. prolifera* at all three photoperiods (L: D of 12:12, 10:14 and 16:8). The different results may be due to differential physiological property between *U. linza* and *U. prolifera*. It has been documented that *U. prolifera* has a higher tolerance to high light intensity compared to *U. linza* (*Cui et al., 2015*). The strong capacity in dealing with high light intensity could contribute to the effect of elevated $CO_2$ and prolonged photoperiod on growth *U. linza* was still positive. Integrating our findings with *Li et al. (2018)* study, the interaction of $CO_2$ and photoperiod on *Ulva* species would be species dependent.

## CONCLUSIONS

This work is the first attempt to clarify the interaction between light/dark and elevated $CO_2$ levels on the physiological responses of *Ulva linza*. We found that the effect of OA on *U. linza* depended on photoperiod. Outbreaks of green tides during spring and summer in China occur when the photoperiod is reaching its peak. Our findings indicate future OA may hinder the occurrence of green tides dominated by *U. linza* in combination with extended photoperiods. More environmental factors, such as temperature and nutrient levels, need to be investigated to obtain a more comprehensive understanding on development of green tides in future oceans.

### Funding

This study was supported by the Natural Science Foundation of Jiangsu Province (Nos. BK20161295), the Six Talents Peaks in Jiangsu Province (JY-086), and the "333" project of Jiangsu Province and Priority Academic Program Development of Jiangsu Higher Education Institutions. The funders had no role in study design, data collection and analysis, decision to publish, or preparation of the manuscript.

## Grant Disclosures

The following grant information was disclosed by the authors:
Natural Science Foundation of Jiangsu Province: BK20161295.
Six Talents Peaks in Jiangsu Province (JY-086).
"333" project of Jiangsu Province and Priority Academic Program Development of Jiangsu Higher Education Institutions.

## Competing Interests

The authors declare there are no competing interests.

## Author Contributions

- Furong Yue conceived and designed the experiments, performed the experiments, analyzed the data, contributed reagents/materials/analysis tools, prepared figures and/or tables, authored or reviewed drafts of the paper, approved the final draft.
- Guang Gao conceived and designed the experiments, analyzed the data, authored or reviewed drafts of the paper, approved the final draft.
- Jing Ma performed the experiments, analyzed the data, authored or reviewed drafts of the paper, approved the final draft.
- Hailong Wu approved the final draft.
- Xinshu Li analyzed the data, contributed reagents/materials/analysis tools, approved the final draft.
- Juntian Xu conceived and designed the experiments, analyzed the data, contributed reagents/materials/analysis tools, authored or reviewed drafts of the paper, approved the final draft.

## Field Study Permissions

The following information was supplied relating to field study approvals (i.e., approving body and any reference numbers):

*U. linza* was collected from the coastal water of Gaogong peninsula (119.3°E, 34.5°N), Lianyungang, Jiangsu Province, China. Gaogong peninsula is a public place and no approval is required for collecting naturally growing Ulva species in China because Ulva species cause green tides in coastal waters of the Yellow Sea.

## Data Availability

The raw data are available in Data S1.

## Supplemental Information

Supplemental information for this article can be found online at http://dx.doi.org/10.7717/peerj.7048#supplemental-information.

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
