# Peer review of "Future CO2-induced seawater acidification mediates the physiological performance of a green alga Ulva linza in different photoperiods"

_PeerJ, doi:10.7717/peerj.7048_

## Round 0.1 · original submission · Minor Revisions

Apart from what is requested by the reviewers, standard practice in ocean acidification experiments is to include details on the manipulation of the carbonate system. The manuscript needs to include this information. Regarding the monitoring of the pH, please specify how you measure it and in which pH scale. Please provide more information on the CO2 instrument you used to manipulate the CO" (HP 1000 G-D, Ruihua Instruments, Wuhan, China).

·

Basic reporting

Needs clarification at many points.

Experimental design

No Comments

Validity of the findings

No comment

Additional comments

This manuscript reports innovative and significant results, but needs significant improvement, as indicated below.

Line 28. Is the 28% 48% of the total anthopogenic CO2 that has been produced, or 48% of what is not in the atmosphere?

Line 33. ‘ocean acidification’, measured as the decrease in surface ocean pH since the pre-industrial Holocene, is not the cause of the perturbations of the inorganic carbon system, but is one of the outcomes of introducing more carbon dioxide, along with an increase in bicarbonate and decrease in dissolved carbonate.

Line 39. Reference needed for the macroalgal contribution to marine primary productivity.

Line 51. Ulva australis is not ‘giant’, nor is it a kelp: a kelp is a member of the Laminariales, or, more broadly, including some members of the Desmarestiales and Fucales. ‘giant kelp’ should be attached to Macrocystis pyrifera.

Lines 58-59. Clarify the positive and negative effects.

Line 62 .’longer’ rather than ‘increased’.

Line 65. ‘increase’ rather than ‘upregulate’; this latter term is best reserved for the changes in mRNA and protein in cells.

Line 71. There is an implication here that green tides were first observed in the Yellow Sea in 2007, rather than earlier elsewhere in the world ocean. Clarify. See, for example, Taylor 1999 Botanical Journal of Scotland 51: 193-203.

Line 74. Ulva (including Enteromorpha) is not the only genus of green tide alga. See Tsutsui et al. 2015 Int Aquat Res 7: 47-62.

Lines 77-78. This single examples contrasts with the effects (increased, no change, decreased) on growth of Ulva species mentioned on lines 43-52.

Line 89. By what criteria (morphological? molecular?) was the alga identified?

Line 91.’Healthy’ by what criteria.

Line 97. ‘adults’ by what criteria?

Line 108. ‘were not obtained’: does this mean there was no variation in fresh weight, or that fresh weights we not determined?

Lines 153-154. Why could not absolute ETR values be determined? It is not difficult to measure the absorptance of Ulva (Longstaff et al. 2002 Photosynthesis Research 74: 281-293).

Line 160. Why were photosynthesis measurements made, as well are those of respiration?

Line 189. What is meant by ‘strong’ in this context?

Lines 199-200. Mention the species of Ulva that were used.

Line 202. ‘algal physiology’ includes a very range of activities. Be more specific.

Lines 203-215. This argument requires that the decreased external pH increases the influx rate of H+ and/or increases the energy cost H+ efflux in maintaining the pH of intracellular compartments.

Reviewer 2 ·

Basic reporting

The article is well written, clear, and concise. However, it is too concise in some parts of the paper and I think there is potential to add more to the discussion section to thoroughly cover all aspects of the results of this experiment. While the grammar is okay overall, I found many small errors. I pointed out some in the notes below, but there are more, and this paper should be edited before resubmission.

Experimental design

The research question of this paper is relevant to understanding how green tide will be affected by future ocean acidification. Many environmental factors can alter the effect of OA on Ulva spp, the primary bloom forming species in the Yellow Sea, including light. Understanding the physiology by doing lab experiments can help us understand what could happen in the natural environment and can contribute to our understanding of algal physiology. This paper collected algae from a natural site and subjected it to three photoperiods and two CO2 levels to understand the effects on growth, chlorophyll fluorescence, respiration, and chlorophyll. The experimental design is appropriate, although some of the methods for the alteration of CO2 could be described in greater detail, particularly the carbonate chemistry.

Validity of the findings

The finding of an interaction between CO2 and photoperiod on the growth of Ulva is valid, but the discussion lacks a comprehensive interpretation of the results. The authors did attempt to connect various physiological responses to the conditions of the experiment and compare it to the results of similar research, but it was superficial and did not explore the details that would help readers understand the meaning of the results in the context of the great body of knowledge.

Additional comments

Abstract

The abstract does a good job at explaining the main findings from the experiment and their significance.

Lines 15-16: Describe how the growth and Yield responses are similar. In the graphs, the patterns are similar in the 8:16 and 16:8 photoperiods, but not the 12:12 photoperiod and the statistical significance of the results differ, so the statement is misleading without clarification.
Line 17: change “Non-fluorescent chemical” to “Non-photochemical”

Introduction

Line 31: change “04” to “0.4”
Lines 33-35: Munday et al. 2011 citation is not in literature cited. The effects of OA on calcareous plankton and especially fishes are not relevant to this study and should be taken out anyways. Check that all cited literature is in the literature cited.
Lines 35-36: change “no-calcareous” to “non-calcareous” and add a citation to this sentence and state why they may benefit.
Line 39: cite the original paper for “macroalgae accounts for 10%...”
Line 41: cite or combine with next sentence
Line 43: Change “concentration” to “concentrating”
Line 43: Explain why the influence of OA on macroalgae is species-specific based on the previous sentences or other references. Before you state that most macroalgae use CCMs, so that does not naturally lead to the conclusion that “Therefore, the influence of OA…is species-specific”. Explain why in greater detail.
Line 43: change “positively” to “positive”
Line 44: change “influences” to “influences on the growth”
Line 51: Remove part about the giant kelp Ulva australis and add before Macrocystis
Line 52: change “reduces” to “reduced”
Lines 55-57: after temperature and nutrients you could add “light” since that is what is paper focuses on. Make sure to include an appropriate citation.
Lines 58-61: It isn’t immediately clear what you mean by positive effects in the daytime and negative impacts during the night. You could expand on this a little by talking about diurnal fluctuations of pH due to photosynthesis and respiration.
Line 63: Elaborate on what you mean by strengthening the control on the CCM
Line 64: You need do work on developing a convincing argument for why the light dark cycle is important in OA. You touched on a few reasons but lack the detail for why it might have an effect. If you expand more on a few points from this paragraph such as CCMs and diurnal fluctuations in pH it would be more convincing.
Lines 64 -66: The sentence about pH upregulation in kelp forests seems random and not within the scope of this paper.
Line 67: By “opportunities” do you mean “growth”?
Lines 68: Which species were tested by Nielsen 1997 and Rost et al. 2003?
Lines 80-83: In your hypothesis you mention the down-regulation of CCMs but did not include any background information or previous studies that show this occurs. You could add that to the previous paragraph when I mentioned you could add more detail.

Materials and Methods

1. What kind of post-hoc analysis was done?
2. Were carbonate chemistry parameters measured and how was pH monitored? What was the pH of the different treatments
3. What kind of curve did you use to fit your RLC? In Fig. 3, it almost looks like there could be photoinhibition occurring at PAR 1000. See Ralph and Gademann 2005 for advice on curve fitting using RLC.

Line 97: change “approximate” to “approximately”
Line 103: How was pH measured to make sure the variation was maintained to between ± 0.05?
Line 105: Its not clear if the experiment lasted 6 days and the measurements were taken on the last 3 days or if the experiment was 9 days and the measurements were taken on the last 3 days.
Lines 107-111: Did you average the RGR across all the 2-day time periods throughout the experiment or only during the last three days as stated previously?
Line 120: change “Non-fluorescent chemical” to “Non-photochemical”
Line 127: How long was the respiration rate measured for?
Line 135: There is what appears to be a spot for a reference that needs to be filled in after SPSS

Results

Line 143: How can elevated CO2 and the light dark cycle both have the greatest effects? This is not clear.
Line 146: be consistent with using L:D or light/dark cycle
Line 152: It doesn’t really look like NPQ increase from 12:12 to 16:8. So be more specific about the results in the results section and describe what happen during each photoperiod.
Line 159: Same comment as above. You need to describe all your results here, not just what generally happened, because the exceptions may be important.

Discussion

1. The discussion seems a little sparse as it stands now. You didn’t mention the results from the Rapid light curve in your discussion.
2. I think it would be important to discuss the findings of Li et al. 2018. Their study was similar to yours but did not find an interacting effect using the species Ulva prolifera. What do you think these differences in results were caused by?
3. I think more thought needs to be put into the discussion by comparing your results to other similar studies and by connecting various physiological mechanisms. This has been done well already, but I think there is room to expand.

Line 182-189: There is a lot of variation in how photoperiod affects different species which is described here. Is there life history features of the different types of algae that cause it to have different optimal growths at different photoperiods? Like are some fast-growing or slow growing or what kind of habitat are they typically found in? Season might play an important role here to. If the algae are collected from a natural environment and then used in a lab experiment, they may still be acclimated to the natural photoperiod. All these aspects could potentially be covered in the discussion, because the discussion at present is very lacking in detail and is only two paragraphs long.
Line 210-212: Its not clear how saved energy from down regulated CCMs could damage the algal photosystem. You should elaborate.

Figures

1. Describe what the lines and letters over the bars represent.
2. Is Fig. 3 necessary? It’s difficult to differentiate between the different treatments, so either make it clear (maybe by separating HC and LC treatments?) or remove it since the information from the graph is summarized in table 3.

---

## Round 0.2 · Minor Revisions

Dear authors,
Thank you for your submission. There is one last issue that needs to be addressed. Figure 5:
1) Please use the same range for Chl a and Chl b y axes.
2) Please double check the letters denoting significant difference, they are very hard to interpret and there might be some typos as well. Maybe it is better to have asterisk instead.

---

## Round 0.3 · accepted · Accept

Dear authors,
I am happy with all the modifications